# Glass-Ceramic Protective Coatings Based on Metallurgical Slag

Alexander V. Gorokhovsky [1,2], Gleb Yu. Yurkov [3], Igor N. Burmistrov [1,*], Angel F. Villalpando-Reyna [4], Denis V. Kuznetsov [1], Alexander A. Gusev [1], Bekzod B. Khaidarov [1], Yuri V. Konyukhov [1], Olga V. Zakharova [1] and Nikolay V. Kiselev [1,5]

1   Department of Functional Nanosystems and High Temperature Materials,
    National University of Science and Technology (MISIS), 119049 Moscow, Russia
2   Department of Chemistry and Chemical Technology, Yuri Gagarin State Technical University of Saratov,
    410054 Saratov, Russia
3   N.N. Semenov Federal Research Center of Chemical Physics, Russian Academy of Sciences,
    119334 Moscow, Russia
4   Department of Engineering Ceramics, The Center for Research and Advanced Studies of the National
    Polytechnic Institute (CINVESTAV) Unidad Saltillo, Ramos Arizpe CP25900, Coahuila, Mexico
5   Engineering Center, Plekhanov Russian University of Economics, 36 Stremyanny Lane,
    117997 Moscow, Russia
*   Correspondence: glas100@yandex.ru

**Abstract:** Pyroxene glass-ceramic enamels based on combinations of blast furnace slag and some additives were produced and investigated. The batch compositions and technological regimes of enameling were developed to produce high temperature protective coatings for carbon steel (ASTM 1010/1008). The composition of raw materials was selected to match the values of the thermal expansion coefficients of the glass-ceramic coating ($\sim 11 \cdot 10^{-6}$ K$^{-1}$) and metal substrate ($\sim 12 \cdot 10^{-6}$ K$^{-1}$) taking into account the temperatures of fluidization ($T_f \sim 800°$) and crystallization ($T_c = 850-1020$ °C) of the corresponding glasses. The covered and thermally treated samples of carbon steel were produced using single-layer enameling technology and investigated to specify structure, phase composition and properties of the coating and coating-steel interface. The obtained coatings were characterized with excellent adhesion to the steel (impact energy $\sim 3$ J) and protective properties. The closed porous structure of the coatings promoted low thermal conductivity ($\sim 1$ W/(m·K)) and high (up to 1000 °C) thermal resistance, whereas the pyroxene-like crystalline phases supported high wear and chemical resistance as well as micro-hardness ($\sim 480$ MPa) and thermal shock resistance (>30 cycles of 23–700 °C). The obtained cheap coatings and effective protective coatings could be used at the temperatures up to 1100 °C in the corrosive atmosphere and under the action of abrasive particles.

**Keywords:** glass-ceramics; carbon steel; protective coating; structure; properties

## 1. Introduction

Modern engineering techniques demand the creation of new coatings, which must have high efficiency under the action of temperature and an aggressive environment. Among protective ceramic coating systems for industrial and engineering applications, glass–ceramic coatings have advantages of chemical resistance, high temperature stability and superior mechanical properties such as abrasion, impact etc., as compared to other coating materials applied by different forms of thermal spraying (physical and chemical vapor deposition (PVD and CVD), plasma, etc.) [1]. In addition, the ceramic coatings produced by sputtering technologies [2–5] usually are brittle, expensive, and difficult to deposit with low-cost processes.

The sol-gel process enables the synthesis of ceramic coating on the steel surfaces with a protective layer based on $SiO_2$, $ZrO_2$, $Al_2O_3$, $TiO_2$. These materials, easily deposited on surfaces with inexpensive processes, showed excellent chemical stability, and improve the

corrosion resistance of metal substrates at low temperatures [6], however, their thermal refractory properties are not high enough. Besides imparting required functional properties such as heat, abrasion and corrosion resistance to suit particular end use requirements, the glass-ceramic coatings in general also provide good adherence, defect free surface and refractoriness [1,7]. However, most of the glass-ceramic coatings described in the literature were developed for stainless steel [8]. The data on protective glass-ceramic coatings recommended for carbon steels are not so extensive.

Some glass-ceramic compositions based on the system of $SiO_2$-$Al_2O_3$-$CaO$-$MgO$-$Fe_2O_3$-$MnO_2$-$K_2O$-$Na_2O$ were developed by Zubekhin and coauthors to coat carbon steels [9,10]. A wear resistant glass-ceramic coating system based on SiO2-B2O3-Al2O3 glass was reported by [11]. A glass-ceramic coating with quartz additions has been developed in [12], as a single coat without prior chemical treatment of the surface, by using the dipping technique on low carbon alloyed steel. A number of $TiO_2$ and $P_2O_5$ nucleated glass-ceramic coating compositions in the system of RO-R'$_2$O-$Al_2O_3$-$SiO_2$ (R = Ca, Mg; R' = Na, K, Li) have been studied for application on various grades of steel and alloy, including mild steel, with an aim of protecting them against mechanical wear and chemical corrosion [13]. A novel environmental barrier double-layer coating system for mild steel consisting of a perhydropolysilazane bond coat and a polysilazane-based glass/ceramic composite topcoat has been developed in [14]. An anti-fouling ceramic coating was developed and applied to carbon steel in the work [15]. Anti-fouling testing and thermal conductivity measurements were performed to evaluate the performance of this coating. However, the modern industry requires that the coatings must have greater efficiency under aggressive environments and can be used in the thermal shock conditions.

Here, it is necessary to note that glass-ceramic materials based on various combinations of industrial waste have been intensively investigated in the last decades and indicated excellent wear, chemical and thermal resistance [16–21]. Among them, the pyroxene-type glass-ceramic materials are very attractive, due to their excellent exploitation properties such as chemical, thermal and wear resistance [21,22].

We proposed that the glass compositions based on industrial waste such as blast furnace slag, fly ash and neutralized sludge of nickel electroplating could represent the glass-forming systems with the chemical composition very similar to that required for the formation of pyroxene-like glass compositions. It is also important that possible varying of the above-mentioned industrial wastes practically does not influence the crystallization behavior of the pyroxene-type glasses due to varied chemical composition of the pyroxen-like solid solutions [22].

In this regard, the aim of this study is to develop a pyroxene-type glass-ceramic coating based on the combination of different industrial wastes using the traditional enamel fritting technique applied to common carbon steel (ASTM 1010/1008) substrates. It is proposed that such coatings could be formed using the single-layer enameling technology (1 coat–1 firing), which minimizes the consumption of raw materials and energy resources due to decreased number of the technological operations. The protective properties of the obtained coatings were investigated taking into account their hardness, wear, chemical and thermal-shock resistance as well as their thermal conductivity.

## 2. Materials and Methods

### 2.1. Design of Glass Composition

The pyroxene-like glass-ceramic materials could be obtained using the raw material mixtures with the chemical composition varied in the range (wt.%): $SiO_2$ (35–60), $Al_2O_3$ (2–15) 3, $Fe_2O_3$ (1–26), % CaO (9–25), MgO (1–20), $R_2O$ (0–12) [22]. Taking into account the gradual global transition from non-renewable raw materials to renewable (plant-based) ones has intensified, the industrial wastes combined with some additives were applied to prepare fritted vitreous materials. The chemical compositions of the applied raw materials are reported in Table 1.

**Table 1.** Chemical composition (wt.%) of different industrial wastes and other raw materials used to formulate glass coating composition.

| Oxide | Raw Materials | | | | | | | |
|---|---|---|---|---|---|---|---|---|
| | Fly Ash | Lime Stone Dust | Dolomite Dust | Silica Sand | Soda Ash | Dried Gavanic Slurry | Blast Furnace Slag | Bento-nite |
| $Na_2O$ | 0.6 | 0.5 | - | - | 58.5 | - | - | 2.1 |
| $MgO$ | 0.7 | 1.2 | 20.6 | - | - | - | 11 | 2.1 |
| $Al_2O_3$ | 24.1 | 2.6 | - | - | - | - | 14.5 | 24.2 |
| $SiO_2$ | 59.7 | 6.7 | 0.2 | 98.5 | - | - | 33.0 | 64.1 |
| $K_2O$ | 1.5 | 0.7 | - | - | - | - | - | 0.7 |
| $CaO$ | 4.6 | 86.1 | 32.5 | - | - | 25.6 | 40.0 | 2.2 |
| $TiO_2$ | 1.5 | - | - | - | - | - | - | 0.2 |
| $Fe_2O_3$ | 6.5 | 1.5 | - | - | - | 12.3 | 0.5 | 2.4 |
| $P_2O_5$ | 0.1 | 0.5 | - | - | - | - | - | - |
| $Cr_2O_3$ | - | - | - | - | - | 1.1 | - | - |
| $NiO$ | - | - | - | - | - | 37.7 | - | - |
| $SO_3$ | - | - | - | - | - | 21.5 | - | - |

These raw materials allow preparing the pyroxene-type glasses with their various combinations. In any case, the blast furnace slag can be considered as the main component of the batch due to its high content of the oxides which participate in the vitrification processes and crystallization of pyroxenes ($CaO$, $MgO$, $Al_2O_3$, $Fe_2O_3$, $SiO_2$). The admixtures of fly ash, silica sand, dolomite and limestone dusts (wastes of the crushed stone production) were selected to optimize the glass composition for the following crystallization, whereas soda ash admixtures were used to regulate the glass transition and fluidization processes ($T_g$, $T_f$).

It is known that the pyroxene-like glasses have a trend of the surface crystallization in the temperature range of enameling (700–900 °C), however, controlling the nucleation and crystal growth rates allows obtaining the glass-ceramic materials, which have high contents of crystalline phases in the glass matrix, using $NiO$ and $Cr_2O_3$ as nucleating agents [23]. That is why, the dried slurry obtained by neutralization of liquid wastes of the nickel electroplating and characterized with high content of nickel and chromium oxides, was introduced in the raw material mixtures too. In addition, it is necessary to note that a presence of $NiO$ promotes wetting of the steel surface by silicate melts [24].

The batch compositions reported in Table 2 were considered to meet the pyroxene-like chemical composition requirements for the glass destined for the protective coating. Table 3 presents the theoretical chemical compositions of the glasses based on these raw material mixtures. The industrial wastes produced in the plants of Severstal Inc. (Cherepovets, Russia) were used here as raw materials.

**Table 2.** Raw material mixtures used to formulate glass composition.

| Raw Material | Batch Number | | | | |
|---|---|---|---|---|---|
| | 1 | 2 | 3 | 4 | 5 |
| Fly Ash | 30 | - | 30 | - | - |
| Limestone | 38 | - | - | - | - |
| Silica Sand | 15 | 23 | 17 | 24 | 24 |
| Galvani Slurry | 7 | 12 | 7 | 13 | 14 |
| Soda Ash | 10 | 8 | 8 | 13 | 13 |
| Dolomite dust | - | 17 | 38 | 20 | 19 |
| Slag | - | 40 | - | 30 | 30 |

**Table 3.** Theoretical chemical composition of the glasses based on the formulated raw material mixtures (Table 2) and their calculated (theor.) and measured (exp.) CTE values.

| Oxide | Number of Glass Composition (Corresponding Batch) | | | | |
|---|---|---|---|---|---|
| | 1 | 2 | 3 | 4 | 5 |
| $Na_2O$ | 7.5 | 5.1 | 6.3 | 9.2 | 9.9 |
| MgO | 0.8 | 8.6 | 10.6 | 8.9 | 8.6 |
| $Al_2O_3$ | 16.3 | 11.7 | 10.6 | 5.2 | 5.5 |
| $SiO_2$ | 46 | 39 | 44.1 | 40.4 | 41.3 |
| CaO | 26 | 27 | 19.8 | 26.4 | 26.0 |
| $TiO_2$ | 0.5 | - | 0.5 | - | - |
| $Fe_2O_3$ | 4 | 1.8 | 2.6 | 2.1 | 2.3 |
| NiO | 3.2 | 6.2 | 2.5 | 7.5 | 6.4 |
| $K_2O$ | 0.7 | - | 0.5 | - | - |
| $Cr_2O_3$ | 0.1 | 0.1 | 0.1 | 0.2 | 0.2 |
| CTE (theor.) $\times 10^6\ K^{-1}$ | 10 | 8.6 | 8.3 | 10.6 | 11.5 |
| CTE (exp.) | $9.7 \pm 0.2$ | $8.6 \pm 0.2$ | $8.4 \pm 0.1$ | $10.3 \pm 0.3$ | $11.6 \pm 0.2$ |

It was assumed that cracking and peeling defects would not occur in the coatings during their formation as when they were exposed to thermal shock and aggressive environment, these coatings will be characterized with optimal values of the coefficient of the thermal expansion (CET). That is why the theoretical CET values were calculated in accordance with [25] for each formulated glass composition to guarantee the compatibility of the coating and the substrate over the temperature range of the expected applications.

The steel ASTM 1010/1008 was selected as a substrate taking into account that this type of carbon steel is used for a wide variety of applications that need high strength and toughness in the form of plates, sheets, bars, and tubes.

The thermo-mechanical behavior of all the investigated materials was determined by dilatometry (Netzsch DIL 402 PC equipment, Selb, Germany) in the temperature range from 20 to 800 °C with a rate of 3 K·min$^{-1}$ using the samples of glass (glass-ceramics) and steel (substrate) of square parallel bases with 6 mm per side and 40 mm length. It was determined that the CTE value of the metal substrate varied from $12.2 \cdot 10^{-6}\ K^{-1}$ (in the range of 20–100 °C) to $15 \cdot 10^{-6}\ K^{-1}$ (in the range of 20–800 °C). That is why, the batch, which allows producing the glass composition No 5 characterized with CTE$_{theor.}$ < CTE$_{steel}$ but closer to CTE$_{steel}$ (Table 3), was chosen for the following experiments, taking into account traditional requirements of the enameling [25,26]. Some preliminary experiments were conducted on the fusion and crystallization of the glass compositions based on the batches No 1–5 and confirmed the correctness of the choice of the batch No 5 as the optimal composition in terms of the thermal expansion properties (experimental CTE values).

### 2.2. Glass Frits and Raw Material Mixtures

The selected batch No 5 was dry homogenized in the rotating drum and melted in alumina crucibles at 1450 °C for 2 h using a Lindberg-Blue high-temperature electric furnace BF51433. To produce a glass frit, the glass-forming melt was poured onto a stainless-steel plate, in order to temper the melt and prevent surface crystallization, and further was ground in a two-station Retsch PM400 planetary mill for 30 min. The glass frit was screened until passing the #325 mesh, considering that this particle size favors green coating of the metal substrates [25,27,28].

The temperatures of vitrification ($T_g$) and crystallization ($T_c$) of the parent glass was determined using the DTA technique (calorimeter Perkin Elmer DTA7), whereas the temperature of fluidization ($T_f$) was estimated taking into account the data of dilatometry too. These results were applied to select the temperature of firing which allowed the formation of the protective glass-ceramic coating by the single-layer enameling technique. This technique is especially promising due to the decreased number of coating operations and

firings. A single-layer enameling prevents warping of thin and large-size metal substrates; the coating becomes more elastic and impact resistant and has reduced thickness.

To form the glass-ceramic coatings, the glass frit (95 wt.%) was dry mixed with powdered bentonite clay (5 wt.%). The obtained mixture was used to produce a slurry (dispersion) prepared in accordance with [24,25,27] an aqueous solution (weight ratio of 1:2) contained 5 wt.% of $Na_2B_4O_7$ (purity of 99%, MosReactiv) and phenol sulfonic acid dispersant (Tamol, BASF) (0.5%). Subsequently, this dispersion was magnetically stirred for 30 min and matured for 24 h (stabilization).

Three slurries with different water contents (40, 50 and 60 wt.%) were used to form green coatings of different thicknesses.

### 2.3. Pretreatment and Coating of the Substrate

Rectangular specimens of $25 \times 50 \times 5$ mm$^3$ were used as a substrate and prepared by cutting the commercial steel plate (ASTM 1008/1010), containing 1.87 wt.% of carbon. To obtain the surface characteristics of a substrate required for the enameling, a chemical treatment of the steel specimens was applied to remove the adhering oxides and surface grease.

The chemical treatment process consisted of immersing the steel specimens in the acidic solution composed of 80 mL of 1 M $H_2SO_4$ and 7.2 g of NaCl [25]. The specimens were introduced in this solution heated up to 80 °C for 15 min. Subsequently, the treated samples were washed in hot water at 95 °C for 1 min to remove residues from the acid solution. After that, the treated specimens were immersed in a solution formed by $Na_2CO_3$ (5 g/L) and $Na_3PO_4$ (3 g/L) at 60 °C for 6 min to inhibit the previous acidic action. Finally, each specimen was dried at 90 °C for 20 min in an oven.

Each specimen of the obtained metal substrate was coated by immersion in the dispersion based on fritted glass. The obtained green coating was dried in an oven at 90 °C for 30 min and further thermally treated at the temperature selected for firing (820, 850 °C).

The muffle furnace was preheated up to 40 °C above the required temperature; subsequently, the green coated specimens were introduced into the hot furnace. Additionally, an evaluation of the time required for the formation of homogeneous coating was performed. It was determined that the burning time of 5 min ensured the total fusion of the green coating and made it possible to avoid generating any cracks or detachment of the coating during the cooling.

### 2.4. Microstructure Analysis and Materials Characterization

The microstructure and chemical composition of the protective layer, obtained after coating and firing, was investigated by scanning electron microscopy (Philips XL30ESEM, SEMTech Solutions, Inc., North Billerica, MA USA) equipped with an energy dispersive spectrometer (EDS, EDAX Pegasus, EDAX, LLC, Pleasanton, CA USA) conducted at 20 kV.

The phase composition of the bulk glass-ceramics and coatings obtained onto the steel surface was carried out using a Philips PW3040 difractometer (CuK$\alpha$ radiation with a nickel filter operating 40 kV and 100 mA). The reflection positions and relative intensities of the XRD patterns were compared to the catalog of the International Center for Diffraction Data (ICDD-2008).

The chemical resistance of the glass-ceramic material used to produce the coatings was estimated by the following standards based on measurement of weight losses:

- water resistance (IRS-3502, Japan): 2 g of the powdered glass-ceramics (fraction of 0.5–0.8 mm) was treated in 50 mL of distilled water for 5 h, filtered and dried for 12 h and weighed;
- chemical resistance to the action of basic and acidic aqueous solutions (GOST 10134-62, Russia): 10 g of powdered glass-ceramics (fraction of 0.5–0.8 mm) were treated in 100 mL of 1 M NaOH or 1 M HCl solutions, respectively, for 3 h at 96 °C; filtered and dried for 12 h and weighed.

Microhardeness of the coatings was measured in accordance with the standard ASTM E384-99 (Tukon microhardness tester, Vickers indenter) using an indentation load of 50 g for 20 s.

To measure the thermal shock resistance of the coating, six coated specimens were heated in an electrical furnace up to 700 °C and then immersed in water at 23 °C (heating–cooling cycle). The number of cycles required to cause any failure was recorded for each specimen.

To estimate wear resistance (wear loss in mg·cm$^2$) we used a modification of the ASTM abrasion standard (G65, by sand between a specimen and a rubber wheel tested for 50,000 revolutions with a rate of $200 \pm 10$ rpm at a load of 45 N) for coating evaluation to rank the relative abrasion resistance and compare the abrasion resistance with that of the uncoated steel surface.

The thermal endurance was estimated by standing the coated steel specimens at 1000 °C with heating and cooling rates of 10 K/min and holding times of 30 min. The thermal conductivity was measured at room temperature and 700 °C in air by the Hot Disk Transient Plan Source (TPS) method according to ISO 22007-2.

The coating-substrate adhesion was estimated in accordance with the Russian standard GOST 24788-2018 which determines minimal acceptable impact energy of the enamel protective coating as 0.6 J. The impact strength (impact energy) test of the coating was carried out on flat areas of the coated metal substrate by the action of a steel ball freely falling from a certain height. After the impact, the surface was tested to recognize any cracking or chipping of the coating. A value of the impact strength was determined as the impact energy which favored delamination of any part of the protective coating from the metal substrate.

## 3. Results

### 3.1. Chemical and Phase Composition (Frits and Coatings)

#### 3.1.1. Chemical Composition

The chemical composition of glass No 5 used as a frit to produce the coatings, in accordance with the data of energy-dispersive X-ray spectroscopy (EDS) and inductively coupled plasma (ICP) analysis, included (wt.%): $Na_2O$ (8.9), MgO (7.8), $Al_2O_3$ (7.9), $SiO_2$ (41.9), CaO (24.9), $Fe_2O_3$ (2.5), NiO (5.9), $K_2O$ (0.2), $Cr_2O_3$ (0.2).

#### 3.1.2. Thermal Behavior

The DSC data obtained for the coarse (0.6–0.8 mm) and thin (0.1–0.16 mm) powders of the parent glass (No 5) are reported in Figure 1a. It is possible to note that the temperature of glass transition of the investigated composition Tg~720 °C and temperature of fluidization is about 800 °C. At the same time, the exothermal peaks of the surface and bulk crystallization appear in the range of 850 and 1020 °C, respectively. The main crystallization process for the coarse powder takes place at T > 1000 °C, while the surface crystallization dominates for the thin powder used to produce glass frit. Taking into account these data, the temperatures 820 and 850 °C have been selected for firing the protective coatings based on the fritted glass No 5. The temperatures of 820–850 °C could allow fluidization as crystallization processes in the glass-forming composition. The temperatures above 850 °C are not recommended for carbon steel enameling [24,25].

The XRD analysis data obtained for the coatings produced at 820 and 850 °C are reported in Figure 1b. A firing at 820 °C did not allow obtaining the glass-ceramic coating, whereas at 850 °C crystallization processes take place in the system investigated; that is why the following experiments were conducted using a firing of green coatings at 850 °C.

### 3.2. Structure

The microphotographs of the obtained coatings are reported in Figure 2. The structure of the external surface for the coating fired at 850 °C is well crystallized (Figure 2b),

continuous and homogeneous, and has some inclusions of open pores. The structure of the coating formed at 820 °C is more rough and inhomogeneous (Figure 2c).

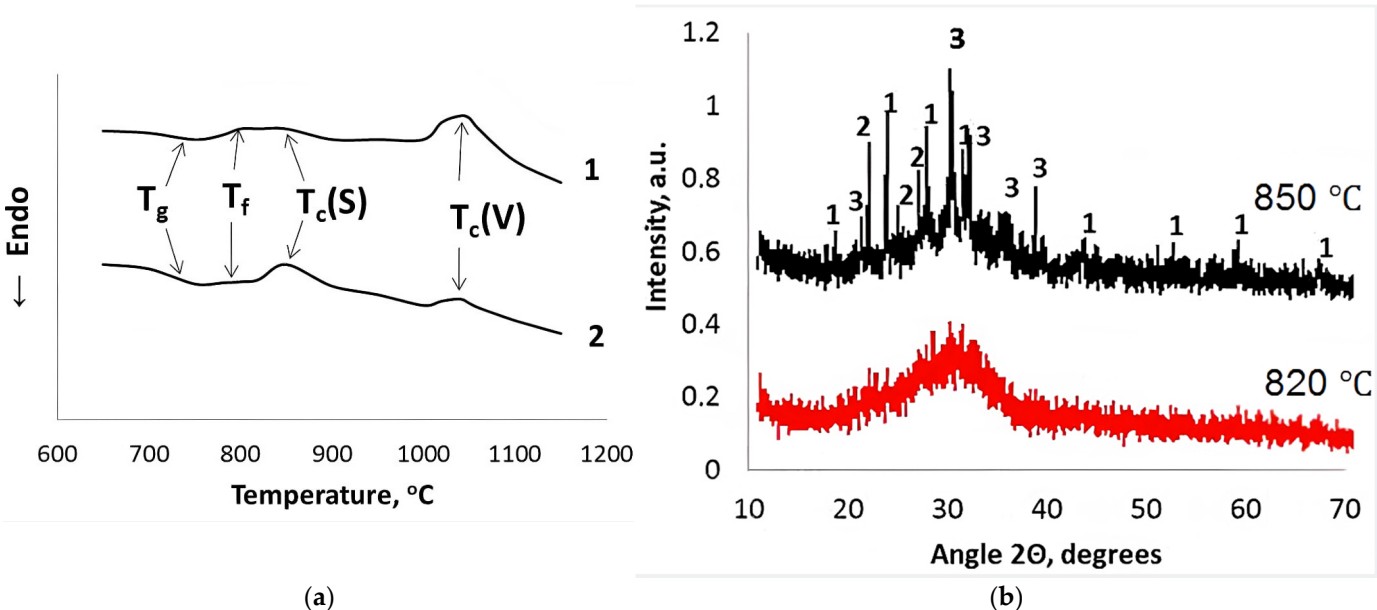

(**a**)  (**b**)

**Figure 1.** DSC data obtained for the coarse (1) and thin (2) powders of glass frit (**a**) and XRD patterns of the coarse glass powder fired at 820 and 850 °C (**b**). 1—nepheline, 2—gehlenite, 3—pyroxene. $T_g$, $T_f$, $T_c(S)$ and $T_c(V)$–the temperatures of glass transition, fluidization, surface and bulk crystallization, respectively.

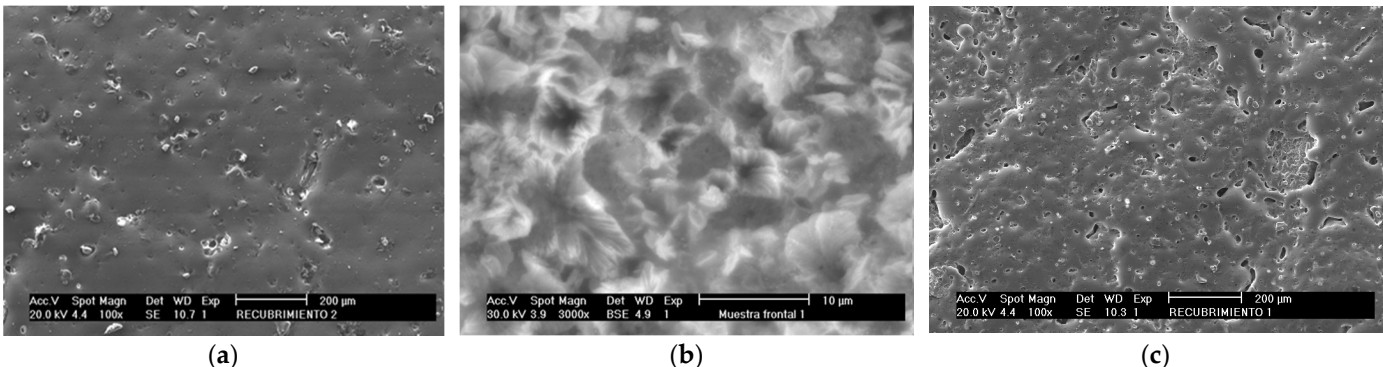

(**a**)  (**b**)  (**c**)

**Figure 2.** Microphotographs (SEM) of the protective coating fired at 850 °C (**a**,**b**) and at 820 °C (**c**).

The cross-section microphotographs (Figure 3) indicate that the obtained coating has a glassy matrix and contains closed pores of a different diameter. The size of pores increases in the direction of the external surface, while, near the coating-metal interface these pores almost disappear. Nepheline ($NaAlSiO_4$), gehlenite ($Ca_2Al_2SiO_7$) and pyroxenes (Ca, Mg, Fe, Al) $Si_2O_6$ represent the crystalline phases of the coating. The pyroxen-like crystals form the main crystalline phase in accordance with the XRD data (Figure 1b). It is important that the crystals are mainly formed either on the coating-steel interface or on the surface of closed pores (Figure 3) and on the external surface of the coating (Figure 2b).

The coating fired at 820 °C is characterized with a very low content of the crystalline phases (Figure 3b) and contains pores near the coating-steel interface (Figure 3b).

It is necessary to note that for the coarse glass powder treated at 820 °C the XRD patterns do not present any reflections of crystalline phases (Figure 1b), whereas, in the structure of the protective coating based on thin glass powder with some additives (clay,

borax), several pores demonstrate initial stages of the surface crystallization (Figure 3b), although in the most part of these pores a surface is free of crystals (Figure 3c).

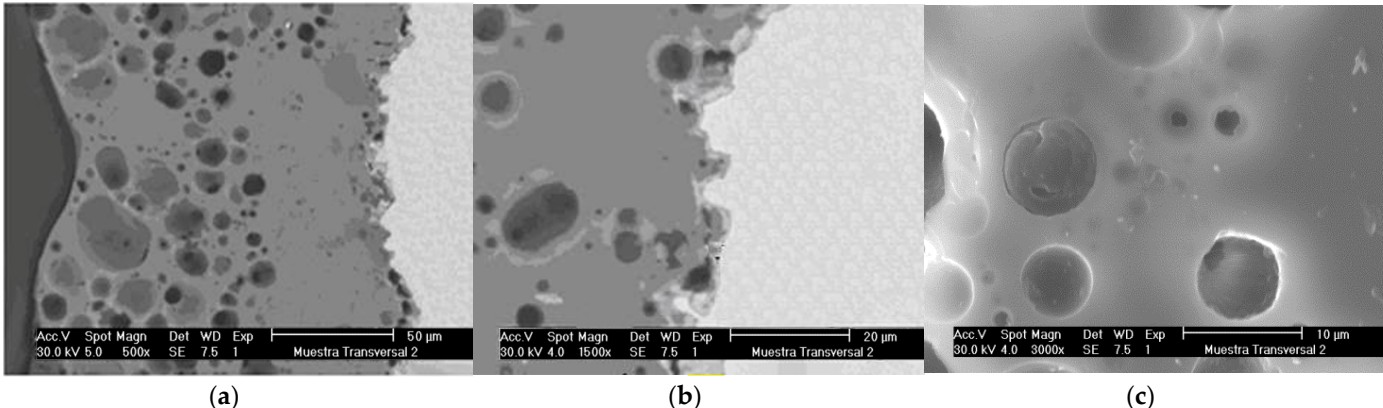

Figure 3. Cross-section micrographs of the coatings produced at 850 °C (a) and 820 °C (b,c).

The data of the EDS point analysis (Figure 4, Table 4) indicate a presence of the chemical elements of the substrate and fritted glass.

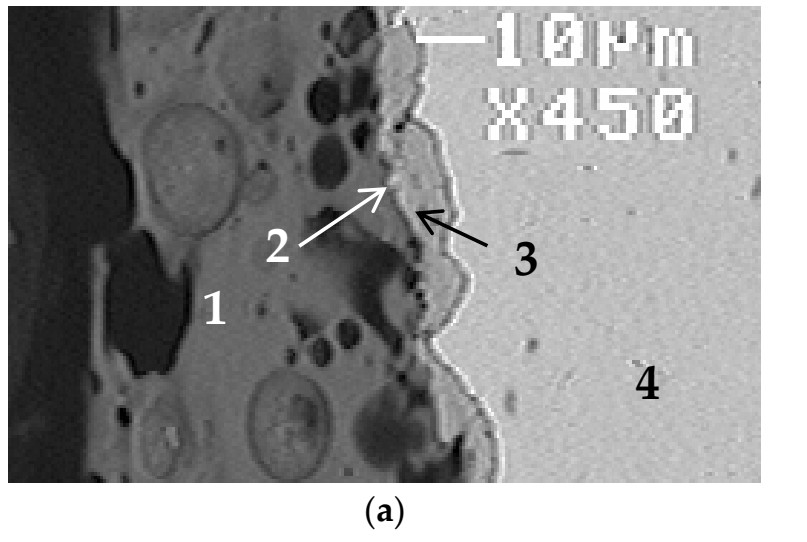
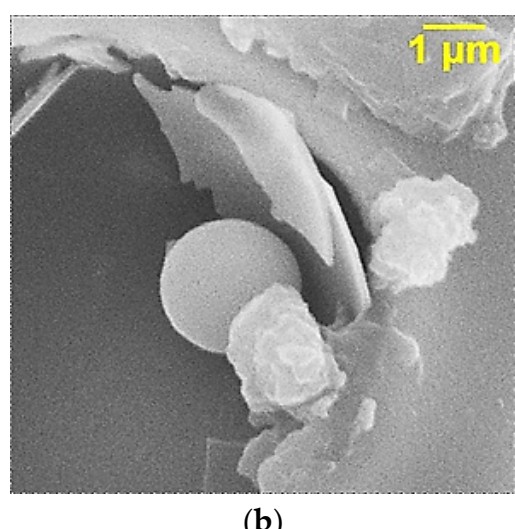

(a)  (b)

**Figure 4.** SEM images of the coating-steel interface formed at 820 °C (**a**) and the trace from a ball impact (6 J) on the surface of the coating obtained at 850 °C (**b**), the elemental composition corresponding to points 1-4 is shown in Table 4.

**Table 4.** Chemical composition of the metal substrate, different zones of the interface and coating (marked in Figure 4). Data of the EDS point analysis.

| Numer of the EDS Point Analysis | Content of the Chemical Elements, at.% | | | | | | | | |
|---|---|---|---|---|---|---|---|---|---|
| | Si | Ca | Fe | Ni | Na | Mg | Al | O | C |
| 1 | 23.5 | 14.9 | 0.6 | 3.0 | 4.7 | 6.5 | 1.9 | 44.9 | - |
| 2 | 2.2 | 1.5 | 54.8 | 3.1 | 0.2 | - | 0.5 | 37.7 | - |
| 3 | 0.9 | 1.0 | 64.6 | 3.3 | - | - | - | 30.7 | - |
| 4 | - | - | 99.1 | - | - | - | - | - | 0.9 |

### 3.3. Properties

The steel ball impacts (adhesion tests) indicated high adhesion of the coating to the steel surface. The impacts with the energy value of 1 J only promoted a cracking of walls of

some pores located near the external surface of the coating (Figure 4b) and did not cause delamination of the coating.

The properties characterizing other protective functions of the resulting coatings are given in Table 5 and considered in the Discussion.

**Table 5.** Some properties of the protective coatings fired at 850 °C.

| Content of H$_2$O in the Slurry, % | Properties | | | | | | | | | |
| | Thickness, µm | | Weight Losses in Aqueous Solutions, % | | | Microhardness, MPa | ** Relative Wear Resistance, % | *** Thermal Shock Resistance | Impact Energy, J | Thermal Conductivity at 700 °C, W/(m·K) |
| | Coating | * Interface Layer | H$_2$O | Acidic | Alkali | | | | | |
| 40 | 150–180 | 4–10 | 0 | 0.5 | 6.3 | 440 ± 32 | 122 | 22 | 3.1 | 1.0 |
| 50 | 110–135 | 3–8 | 0 | 0.2 | 4.8 | 459 ± 20 | 132 | 25 | 3.3 | 0.8 |
| 60 | 80–110 | 2–5 | 0 | 0.1 | 5.0 | 481 ± 21 | >30 | 154 | 3.6 | 1.1 |

* glass-ceramic coating–metal substrate, ** comparison with the uncoated steel, *** quantity of the thermal cycles 23–700 °C.

## 4. Discussion

Thus, the glass-ceramic coatings, produced using the combination of blast furnace slag, dried galvanic slurry of the nickel electroplating and some additives have excellent protective properties. They meet the standard requirements (GOST 24788-2018) related to the impact strength (impact energy) and are characterized with a good adhesion to the carbon steel surfaces. Impact energy values were higher than 0.6 J and reached 3.1–3.6 J, whereas such characteristics reported in the literature ranged between 1.1 and2.4 J [1,13,29].

In comparison with similar protective glass-ceramic coatings proposed earlier for the carbon steels [9–15,27–30], the obtained coating is characterized with some very attractive exploitation properties.

The experimental data indicate that the developed protective coating has excellent chemical resistance in water and alkali aqueous solutions and acceptable resistance under the action of strong acids.

Abrasion resistance of the coating (1.6 ± 0.1 mg/cm$^2$) increased in 54%, in comparison with the uncoated parent steel surface, and was higher than one reported in previous works (2–3 mg/cm$^2$ after 50,000 cycles of RWAT [1,12]). The hardness of the obtained coating is slightly less of the best characteristics for the analogs (480 MPa and 500–600 MPa [9–11], respectively), however, it is acceptable taking into account high wear resistance and corresponds to the data obtained for the bulk pyroxene-type glass-ceramics [20–22,30].

The thermal conductivity of the coating is also much better for this one mentioned for the analogs (1.1 and 2.6 W/(m·K) [15], respectively).

Spherical pores of various diameters located in the external part of the coating can be considered as a filler of the glassy matrix. In fact, the obtained coating is filled with ceramic porous particles characterized with high mechanical strength and low thermal conductivity. Such structural features of the coating also promote high mechanical properties, thermal and thermal-shock resistance.

Good adhesion of the coating to a surface of the carbon steel substrate and low thermal conductivity promote an excellent thermal shock resistance (more than 30 cycles of 23–700 °C, in comparison with 15–20 cycles of 23–400 °C [1,13]).

The boiling water resistance (100%) as well as acid (>99%) and alkali (~95%) resistance of the obtained coatings are better for the analogs described in the literature (98%–99%, 95%–98%, 92%–94%, respectively, [1,9,10,12,13,31]). The cause of this phenomenon can be explained by dominant surface crystallization of the pyroxene-like glass compositions. As a result, a presence of the chemically resistant ceramic layers as onto the external surface of the coating as on the surfaces of pores, provide improved chemical durability.

It is necessary to note that a thickness of the glass-ceramic coating based on metallurgical slag can be regulated with the H$_2$O contents in the slurry used to form green coating by dipping (Table 4). The obtained results allow one to suggest that an increase

in the viscosity of the slurry based on fritted enamel increases a thickness of the film due to the substrate-coating interaction. However, more thick coatings are characterized with worse thermal shock and wear resistances, whereas very thin coatings have worse abrasion resistance and relatively high thermal conductivity. That is why the slurries containing 50–55 wt.% of $H_2O$ could be recommended to produce the glass-ceramic coatings of the proposed composition characterized with improved protective properties.

The following factors influencing protective properties of the proposed coating have to be taken into account.

A presence of about 6 wt.% of NiO in the chemical composition of fritted glass, used to produce the coating, promotes two useful processes.

The first, in accordance with [22,23,28,30], is related to the nucleation of crystalline phases in the molten silicate glasses. However, in spite of a presence of NiO and $Cr_2O_3$ supporting the crystallization of pyroxenes [21], a surface crystallization dominates in the system investigated; a bulk crystallization only takes place at T > 1000 °C (Figure 1a). Nonetheless, the nucleants favor an intensive growth of the pyroxene-like crystals from the steel-coating interface into a volume of the protective coating (Figure 3a), improving the mechanical strength and thermal-shock resistance of the coating.

In addition, a presence of NiO in the glass composition promotes a perfect adhesion between the coating and metal substrate, achieved due to an appearance of the intermediate layer. This layer has a thickness of 2–8 µm (Figures 3 and 4, Table 4) and, in accordance with the EDS point analysis data, contains the components of fritted glass and metal substrate ($SiO_2$, CaO, NiO, $Fe_2O_3$). Thus, it is possible to assume that the phases supporting adhesion, in particular $\alpha$-$Fe_2O_3$ and ferrites ($NiFe_2O_4$), formed during a firing as a result of the redox processes [31].

$$Fe + NiO = Fe_2O_3 + Ni$$

$$NiO + Fe_2O_3 = NiFe_2O_4.$$

The data of the EDS point analysis (Figure 4, Table 4) indicate that interdiffusion occurs at the interface during the firing of green coatings [26]. The interlayer is formed as a result of the metal substrate oxidation and diffusion of the glass components into the $Fe_2O_3$ structure. It is important that Ni has a trend of incorporation into the structure of the interface layer in the metal form; this is indicated by a gradual decrease in the amount of oxygen in the intermediate layer, when moving from the steel surface to the glass-ceramic coating. Such gradient structure of the interlayer prevents micro-cracking of the ductile metal substrate, which recently has been discovered to be dangerous during the destruction of brittle coatings [32]. In our case the interlayer provides perfect adherence of the coating due to its good impact strength and wear resistance.

Thus, the developed glass–ceramic coating material based on industrial wastes, such as metallurgical slag and waste of nickel electroplating, is useful for specialized engineering and industrial applications. A presence of the pyroxene-like crystalline phases, characterized with high toughness, hardness, thermal, wear and chemical resistance promotes improved exploitation properties in comparison with the analogs containing inclusions of quartz, alumina, magnesium aluminium titanate, lithium titanium silicate or sodium silicates [9–15,31]. Due to high mechanical strength and abrasion resistance, the pyroxene-type glass-ceramic coating has a good potential for its applications for the carbon steel constructions which are used in the conditions where an abrasive action accelerates a failure.

It is important to note that the developed coatings can be produced by the simple one-stage technique of enameling. Such a technique allows one to obtain protective coatings onto the surface of large-sized samples of carbon steel. In our case, the ASTM 1008/1010 carbon steel plates with a size of 4 × 150 × 200 mm were coated using the investigated glass-ceramic composition. However, some special equipment has to be developed to coat the samples of a higher scale. This research as well as more detailed investigation of the mechanical and other exploitation properties will be conducted in the near feature taking into account the requirements of the potential consumers.

## 5. Conclusions

The raw material mixtures based on the combination of powdered blast furnace slag and wastes of nickel electroplating as well as some technological additives (silica sand and soda ash) allow producing the ASTM 1008/1010 carbon steel glass-ceramic coating by the simple one-stage enameling technique, using the 50% aqueous dispersions of the fritted pyroxene-type glass powder with admixtures of bentonite clay, borax and surfactant, and with the following thermal treatment at 850 °C.

The obtained coatings have some structural features: (1) well developed gradient coating-steel interface layer formed, most likely by $\alpha$-$Fe_2O_3$, $NiFe_2O_4$, Fe and Ni; (2) closed spherical pores with the ceramic walls, located near the external surface; (3) totally crystallized external surface formed by nepheline ($NaAlSiO_4$), gehlenite ($Ca_2Al_2SiO_7$) and pyroxene-like crystals.

These structural features promote improved adherence, mechanical properties, thermal shock and chemical resistance of the coating, which allow one to recommend them for the carbon steel constructions used in the aggressive conditions of high temperatures, abrasive action and chemically aggressive media.

**Author Contributions:** Conceptualization, A.V.G.; methodology, G.Y.Y. and Y.V.K.; validation, I.N.B. and N.V.K.; investigation, A.F.V.-R., A.A.G., B.B.K. and O.V.Z.; data curation, D.V.K.; writing—original draft preparation, G.Y.Y.; writing—review and editing, A.V.G. All authors have read and agreed to the published version of the manuscript.

**Funding:** This research was partially financially supported by the Ministry of Science and Higher Education of the Russian Federation in the framework of Strategic Academic Leadership Program "Priority 2030", NUST "MISIS" grant No K2-2020-009.

**Institutional Review Board Statement:** Not applicable.

**Informed Consent Statement:** Not applicable.

**Data Availability Statement:** Not applicable.

**Conflicts of Interest:** The authors declare no conflict of interest.

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
