# Peer review of "Glass-Ceramic Protective Coatings Based on Metallurgical Slag"

_coatings, doi:10.3390/coatings13020269_

Round 1

Reviewer 1 Report

All the different parts of this article (from the abstract, introduction to the discussion and conclusion) are scientifically poorly written and cannot be accepted in this journal.

Author Response

The manuscript was comprehensively revised, the main sections were improved, the experimental part and the discussion were expanded, the novelty points were established in accordance with the specific comments of the reviewers.

Reviewer 2 Report

This paper prepared a glass-ceramic protective coating on carbon steel by using blast furnace slag and some additives. The topic is meaningful, but the organization form is too poor and the data system is incomplete.

1. In section 2.1, there are long text to explain the design for the coating and the section of compositions for the coating. However, in the results, only one coating was prepared and compared. So it may be better to simple explain it instead of these long explanations.

2. The quality of figure 1 is too poor.

3. Surface morphology of the coating fiered at 820 C is also suggested to present in Fig. 2.

4. Crystalline products can be also found for the coating formed at 820 C, why was no XRD peak found in Fig. 1b?

5. In results section, data about coating adhesion. impact strength and EDS results were not presented, but they were described in experimental and discussion section.

6. Author stated that no delamination or cracking of the coating after the impact of steel ball. It would be better to present the surface morphology condition after experiment, such as wear test, impact test.

7. It is suggested to indicate the interface layer in cross section images. Why this interface layer can lead to a good adhesion property of the coating? The interface layer should be give better characterization.

Author Response

Thank you very much for your comments.

We have tried to take into account all the recommendations of the Reviewers regarding requested changes and/or additions in our manuscript. All corrections are highlighted in color.

Question

Answer

There are long text to explain the design for the coating and the section of compositions for the coating. However, in the results, only one coating was prepared and compared.

So it may be better to simple explain it instead of these long explanations

From our point of view, such preliminary theoretical analysis is an important part of the paper, which allows justifying the batch composition before the main experiments.

Certainly, we conducted a number of control experiments with other batch compositions, reported in Table 2, which confirmed the correctness of the choice of the main composition No 5.

It seems that detailed description of these experiments is not interesting to readers, however, we have added some explanations at the end of the part 2.2.

Please see lines 145-148.

The quality of figure 1 is too poor.

We have increased the resolution of the XRD patterns and DSC curve. However, it is necessary to take into account that an intensity of the crystallization processes in the coarse glass powder is relatively weak and, that is why, the intensity of reflections in the diffractograms are not strong and the diffraction patterns are typically characterized by a rather noisy appearance.

Surface morphology of the coating fired at 820 C is also suggested to present in Fig. 2.

Done

Crystalline products can be also found for the coating formed at 820 oC, why was no XRD peak found in Fig. 1b

Thank you for this remark. The XRD patterns reported in Fig.1b correspond to the samples obtained as a result of the thermal treatment of the cylindrical glassy samples at different temperatures.

The X-ray diffraction pattern of the coarse glass powder treated at 820 oC does not contain any reflexes of the crystalline phases, while in the coating produced using a fritted glass has shown a presence of weak surface crystallization in some pores. At the same time, in the most part of pores such crystallization is absent. We have introduced the additional micro- photograph (Fig.3c), which confirm this fact.

This effect for the coating based on powdered glasses is evident, taking into account that the pyroxene glasses have a trend of a surface crystallization. Furthermore, a presence of some additives, such as clay, in the fritted composition could support this process (this comment is introduced into the text).

We corrected this mistake in the footnote of Fig.3.

In results section, data about coating adhesion impact strength and EDS results were not presented, but they were described in experimental and discussion section.

We have introduced these data in the part Results.

Please see added Table 4, modified Table 5 and some description on the lines 301-307.

Author stated that no delamination or cracking of the coating after the impact of steel ball. It would be better to present the surface morphology condition after experiment, such as wear test, impact test.

We have introduced the microphotograph of the coating after the impact test (Fig. 4). Unfortunately, the coating after the wear tests were not investigated by SEM.

It is suggested to indicate the interface layer in cross section images. Why this interface layer can lead to a good adhesion property of the coating? The interface layer should be give better characterization.

We have introduced the data on composition of the interface layer in the Discussion and extended the text explaining a role of this layer.

Please, see lines 375-384.

Reviewer 3 Report

The pyroxene glass-ceramic enamels based on combinations of blast furnace slag and

some additives were produced and investigated. In principle the topic is suited for Coatings, however, the manuscript needs to be improved considerably before publication. I state some more specific points, which need improvement, in the following

1.     Does the content of H2O in slurry have any influence on properties of coatings?

2.     The coating-substrate adhesion was estimated in accordance with the Russian standard GOST 24788-2018, and in Line289 “The impact of a steel ball did not cause delamination or cracking of the material of all the investigated glass-ceramic coatings”. So, what does the thermal shock resistance mean?

3.      Recent experimental observations show that the fracture of a brittle coating can cause the micro-cracking of the ductile metal substrates, threatening the safety and reliability of engineering structures(doi.org/10.1016/j.actamat.2018.04.017). Is there any influence of coatings on substrate during thermal shock. And please provide the morphology after thermal shock.

4.     the glass composition No 5 was selected due to the similar CET value compared with metal substrate. One can see that CET value, while important, are not all-determining. And Line 345-347 “In fact, the obtained coating is filled with ceramic porous particles characterized with high mechanical strength and low thermal conductivity. Such structural features of the coating also promote high mechanical properties, thermal and thermal-shock resistance”. Does any samples prepared by other glass compositions were analyzed? 

5.     Does the interdiffusion occur at the interface during preparation or thermal shock? For ceramics coating, the properties of nitrides were influenced by the interface between coating and metal substrate (doi.org/10.1080/09500839.2019.1656351). For the coatings in manuscript, the coating formed by slurry with 60 wt.% water contents achieved good results. Was it caused by the interlayer or coating itself? And some explanations of the experimental results are best described in detail.

Author Response

Thank you very much for your comments.

We have tried to take into account all the recommendations of the Reviewers regarding requested changes and/or additions in our manuscript. All corrections are highlighted in color.

Question

Answer

Does the content of H2O in slurry have any influence on properties of coatings?

Water content influences a quality of the obtained coating due changed viscosity and wettability. Some comments on this question are introduced into the text. Please, see lines 346-354.

The coating-substrate adhesion was estimated in accordance with the Russian standard GOST 24788-2018, and in Line289 “The impact of a steel ball did not cause delamination or cracking of the material of all the investigated glass-ceramic coatings”. So, what does the thermal shock resistance mean?

The thermal shock and adhesion (ball impact) tests are different. In this case the thermal shock resistance is a stability of sample structure under rapidly transient mechanical load caused by a sample temperature change.

We modified the text to clarify this question

Please see lines 219-222, 233-234 and 237-239.

Recent experimental observations show that the fracture of a brittle coating can cause the micro-cracking of the ductile metal substrates, threatening the safety and reliability of engineering structures (doi.org/10.1016/j.actamat.2018.04.017). Is there any influence of coatings on substrate during thermal shock. And please provide the morphology after thermal shock.

A study of the structure and mechanical properties of the metal substrate after exposure to the thermal shock was not included in the objectives of our research. However, this aim is interesting and important, therefore we have described it in the final part of the Discussion.

The thermal shock impact did not influence a coating morphology, however, the specimens, which did not pass the thermal cycling, had cracked and delaminated parts of the coating. Unfortunately, the coating after thermal shock were not investigated by SEM.

the glass composition No 5 was selected due to the similar CET value compared with metal substrate. One can see that CET value, while important, are not all-determining. And Line 345-347 “In fact, the obtained coating is filled with ceramic porous particles characterized with high mechanical strength and low thermal conductivity. Such structural features of the coating also promote high mechanical properties, thermal and thermal-shock resistance”. Does any samples prepared by other glass compositions were analyzed?

We have introduced some additional comments on the factors influencing the properties of the obtained glass-ceramic coating.

Other glass (batch) compositions analyzed in the theoretical part were not used to produce the coatings. However, the chemical composition of all the glass compositions mentioned in Table 2 are close and, taking into account the data obtained in our previous works (references No 20,21, 28 etc.), their crystallization behavior has to be similar.

Does the interdiffusion occur at the interface during preparation or thermal shock?  For ceramics coating, the properties of nitrides were influenced by the interface between coating and metal substrate (doi.org/10.1080/09500839.2019.1656351).

We agree with this comment and have introduced the data on structure and composition of the interface layer in the part Results and extended the text, explaining a role of this layer

Please see fig. 4, table 4 and lines 375-384.

For the coatings in manuscript, the coating formed by slurry with 60 wt.% water contents achieved good results. Was it caused by the interlayer or coating itself? And some explanations of the experimental results are best described in detail.

More detailed explanation of the influence of water content in the slurry on a quality of coating are introduced in the text. Please see lines 348-354.

Reviewer 4 Report

Reviewer # :  The authors reported the interesting results and conducted a significant work. The manuscript was well written and organized. However, there existed several issues that should be revised.

1.     Reformulate the abstract in order to clearly show the strengths of this work.

2.     The novelty of the work should be established.

3.     The experimental part must be detailed.

4.     Why you did not use other more powerful techniques in characterization?

5.     Please provide figures of high resolution.

6.     Comparison with previous works are not reported.

7.     Conclusion should be concise.

Thus, the manuscript should experience the major revision before acceptance.

Author Response

Thank you very much for your comments.

We have tried to take into account all the recommendations of the Reviewers regarding requested changes and/or additions in our manuscript. All corrections are highlighted in color.

Question

Answer

1. Reformulate the abstract in order to clearly show the strengths of this work.

We have tried to improve a quality of the abstract (Please see lines 24-27, 30) and manuscript with more detailed description of the experimental part, novelty etc. All corrections are highlighted in color.

The main novelty points were established in the conclusion on lines 404-418.

2.     The novelty of the work should be established.

3. The experimental part must be detailed.

4. Why you did not use other more powerful techniques in characterization?

This research has technical character and we considered the combination of exploitation properties of the obtained coatings as a main factor. More powerful kinds of the analytical equipment could present some data interesting from the point of view of fundamental research, but would not support the technical characteristics of the obtained and investigated coatings as well as influence their potential application.

Additionally, the EDS analysis and more SEM images were included (Please see fig. 4 and table 4).

5. Please provide figures of high resolution.

Done, where it was possible

6. Comparison with previous works are not reported.

Some comparisons of the obtained results with the data reported in earlier published works take place (look at the pages 9 and 10. Unfortunately, more detailed comparison meets some problems related to different kinds of testing (standards) used in various publications.

7. Conclusion should be concise.

Done

Reviewer 5 Report

The authors presented interesting research results of a new ceramic coating. An important aspect of the novelty is the use of potential production waste as components of the new solution.

On what largest surface were the authors able to apply their coating? Is it a typical research setup at the current stage or is there an idea for applications on large surfaces?

Did the authors, in addition to the typical mechanical and temperature properties of the obtained coatings, test, for example, the tightness of the obtained coating, which is very important when used as protection against an aggressive external environment?

Author Response

Thank you very much for your comments.

We have tried to take into account all the recommendations of the Reviewers regarding requested changes and/or additions in our manuscript. All corrections are highlighted in color.

Question

Answer

On what largest surface were the authors able to apply their coating? Is it a typical research setup at the current stage or is there an idea for applications on large surfaces?

We have introduced the data on a size of the coated metal samples as well as some comments on application of such coatings for large surfaces.

Please see lines 396-399.

Did the authors, in addition to the typical mechanical and temperature properties of the obtained coatings, test, for example, the tightness of the obtained coating, which is very important when used as protection against an aggressive external environment?

Thank you for this interesting question-suggestion. A detailed investigation of the structural features and properties of the obtained coatings in the real exploitation conditions, we assume would be redundant within the framework of this publication, but they are important and will be done in our feature research. Some comments on this aims are introduced in the text.

Round 2

Reviewer 1 Report

This paper reports on the Glass-ceramic protective coatings based on metallurgical slag. The English language needs to be thoroughly reviewed throughout the paper. The article is incomplete in parts, with some explanations and clarifications needed for better understanding. I do recommend that this article be published in Journal of Coatings after Major revision.

1)      The English writing in the paper also needs to be improved as there are many grammatical errors and incomplete sentences throughout the paper that made it difficult to read.

2)      In the abstract section, the main findings of the research should be written numerically and quantitatively.

3)      At the end of the abstract, write a general conclusion.

4)      In the introduction and discussion sections, use the content of the following articles and refer to them in the references section: 10.1515/ijcre-2019-0130; Application of response surface methodology on cefixime removal from aqueous solution by ultrasonic/photooxidation. International journal of pharmacy and technology. 2016;8(3):16728-36.; Synthesis of magnetic Fe3O4/activated carbon prepared from banana peel (BPAC@ Fe3O4) and salvia seed (SSAC@ Fe3O4) and applications in the adsorption of Basic Blue 41 textile dye from aqueous solutions. Applied Water Science. 2022 May;12(5):1-1.

5)      In the introduction section, a number of similar studies should be mentioned.

6)      All abbreviations used in this article should be written in full at the beginning and the abbreviation should be written in parentheses and their abbreviation should be written in the continuation of the article.  

7)      In the Methods, the type of study to be written.

8)      The discussion section of the article is poorly written. Must be upgraded with new articles.

9)      The strengths and weaknesses of this study compared to other studies should be written.

Author Response

No

Suggestions:

Answers:

1

)      The English writing in the paper also needs to be improved as there are many grammatical errors and incomplete sentences throughout the paper that made it difficult to read.

We’ve tried to improve English there it was possible

2

2)      In the abstract section, the main findings of the research should be written numerically and quantitatively.

Done

3

3)      At the end of the abstract, write a general conclusion.

Done

4

4)      In the introduction and discussion sections, use the content of the following articles and refer to them in the references section: 10.1515/ijcre-2019-0130; Application of response surface methodology on cefixime removal from aqueous solution by ultrasonic/photooxidation. International journal of pharmacy and technology. 2016;8(3):16728-36.; Synthesis of magnetic Fe3O4/activated carbon prepared from banana peel (BPAC@ Fe3O4) and salvia seed (SSAC@ Fe3O4) and applications in the adsorption of Basic Blue 41 textile dye from aqueous solutions. Applied Water Science. 2022 May;12(5):1-1.

Sorry, but these papers are so far from the object of our paper and it is difficult to justify citing them here.

5

5)      In the introduction section, a number of similar studies should be mentioned.

We tried to collect all the data published in the literature on glass-ceramic protective coatings to show the state of art

6

6)      All abbreviations used in this article should be written in full at the beginning and the abbreviation should be written in parentheses and their abbreviation should be written in the continuation of the article. 

Done

7

In the Methods, the type of study to be written.

Done

8

8)      The discussion section of the article is poorly written. Must be upgraded with new articles.

From our point of view, the discussion has to be carried out using the results published on glass-ceramic coatings only. The most part of articles on protective coatings published in recent years is related to high temperature refractive coatings for stainless steel. For these systems, the coating process differs significantly from the system considered in our manuscript.

9

9)      The strengths and weaknesses of this study compared to other studies should be written.

It was done in the discussion

Reviewer 2 Report

No more questions. It could be accepted.

Author Response

Thank you for reviewing our paper!

Reviewer 4 Report

I agree with the detailed responses for the authors.

Author Response

Thank you for reviewing our paper!